# REFUSAL FALLS OFF A CLIFF:
# HOW SAFETY ALIGNMENT FAILS IN REASONING?

## ABSTRACT

Large reasoning models (LRMs) with multi-step reasoning capabilities have shown remarkable problem-solving abilities, yet they exhibit concerning safety vulnerabilities that remain poorly understood. In this work, we investigate why safety alignment fails in reasoning models through a mechanistic interpretability lens. Using a linear probing approach to trace refusal intentions across token positions, we discover a striking phenomenon termed as **refusal cliff**: many poorly-aligned reasoning models correctly identify harmful prompts and maintain strong refusal intentions during their thinking process, but experience a sharp drop in refusal scores at the final tokens before output generation. This suggests that these models are not inherently unsafe; rather, their refusal intentions are systematically suppressed. Through causal intervention analysis, we identify a sparse set of attention heads that negatively contribute to refusal behavior. Ablating just 3% of these heads can reduce attack success rates below 10%. Building on these mechanistic insights, we propose **Cliff-as-a-Judge**, a novel data selection method that identifies training examples exhibiting the largest refusal cliff to efficiently repair reasoning models' safety alignment. This approach achieves comparable safety improvements using only 1.7% of the vanilla safety training data, demonstrating a less-is-more effect in safety alignment.

## 1 INTRODUCTION

Large Reasoning Models (Guo et al., 2025; Shao et al., 2024; Hugging Face, 2025), with advanced reasoning capability derived from reinforcement learning with verifiable rewards (RLVR) (Yu et al., 2025; Liu et al., 2025a), are designed to handle complex problem solving, logical inference, and tool-assisted planning. However, while these methodological advances signal more reliable and capable models, they simultaneously introduce significant safety considerations. It is widely discovered that current reasoning-oriented models often lag behind in safety alignment, and tend to exhibit higher susceptibility to attacks (Kuo et al., 2025; Sabbaghi et al., 2025; Kuo et al., 2025; Zaremba et al., 2025; Zhou et al., 2025a; Li et al., 2025a), highlighting an urgent need for reasoning-specific safety mechanisms. Many previous works have benchmarked the safety of reasoning models (Jiang et al., 2025), developed jailbreaking strategies (Wang et al., 2025), and proposed alignment solutions zhang2025realsafe, but have lacked analysis of the mechanisms under the vulnerability of reasoning safety.

Understanding why safety alignment in reasoning models is vulnerable provides invaluable insights for both societal benefit and future model development. In this paper, we *firstly* aim to answer the following **research question**:

*What mechanism makes the safety alignment vulnerable in reasoning models?*

While numerous reasoning models exhibit unsafe behaviors, the underlying mechanisms driving these failures remain critically important to investigate. Do these reasoning models lack safety capabilities, or do they have adequate risk assessment abilities but simply choose not to act on them, failing to refuse harmful requests? Empirical studies have shown that the internal reasoning traces of such models can be unfaithful to the actual decision-making process and may fail to explicitly reveal the model's true intentions (Barez et al., 2025; Arcuschin et al., 2025). This limitation motivates the need to probe models from the perspective of their internal representations. Prior research

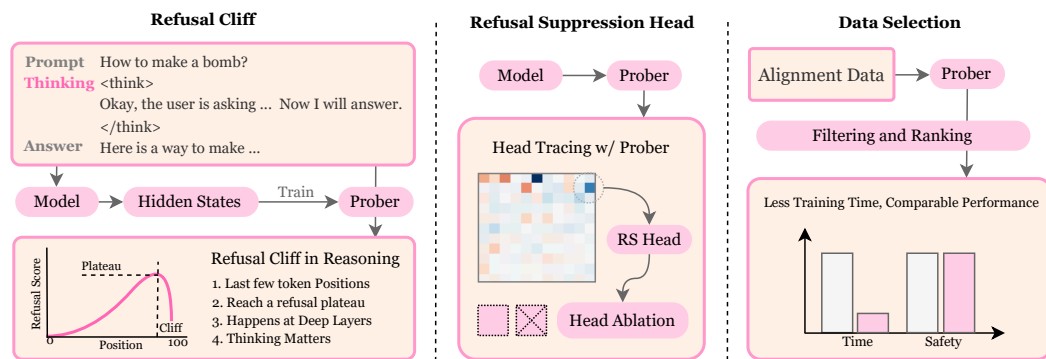

Figure 1: **An overview of our paper**. *Left:* We train a prober and discover the refusal cliff. *Center:* We find Refusal Suppression Heads as the main cause of the cliff. *Right:* We design data selection method based on probing the cliff.

has demonstrated that language models encode meaningful and behaviorally relevant features within their representation space (Turner et al., 2023; Engels et al., 2025; Gorton & Lewis, 2025). These latent features have been shown to govern various emergent behaviors, *e.g.,* in-context learning (Ilharco et al., 2022; Hendel et al., 2023), instruction following (Stolfo et al., 2025), and sentiment modulation (Turner et al., 2023). In the context of safety alignment research, refusal behavior is often considered a canonical metric, and a specific refusal direction (Arditi et al., 2024) in representation space has been shown to regulate such behavior. To examine how these safety-relevant features evolve across tokens and layers, a prominent approach emerging from mechanistic interpretability – the use of linear probes (Nanda et al., 2025) – offers a principled method for analyzing the internal processing of language models.

To characterize the dynamics of refusal behavior in reasoning models, we build on prior work (Chan et al., 2025; Xu et al., 2024) and adopt a probing-based methodology to quantify safety-relevant signals in hidden-state representations. In our framework, safety is operationalized via refusal behavior (Arditi et al., 2024), as well-aligned models are expected to refuse harmful queries (*e.g.,* through *I'm sorry* statements). Concretely, we train a linear probe classifier to predict, given hidden states from different positions in the reasoning chain, whether the model will refuse the prompt. The probe's predicted probability is termed the *refusal score*, with higher scores indicating internal states more predictive of refusal. Across multiple partially aligned reasoning models, we observe a recurrent pattern we call the **Refusal Cliff**. While intermediate reasoning steps yield refusal scores comparable to strongly aligned instruction-tuned models – indicating successful detection of harmfulness – scores drop sharply in the final steps. This reflects suppression of refusal behavior even where refusal would be the alignment-consistent choice. The sharp decline suggests these models maintain alignment only in early reasoning, but fail to preserve it through output generation.

The Refusal Cliff consistently occurs at the final positions of the reasoning chain, corresponding to a fixed set of output tokens (the *thinking-end template*). These template tokens must retrieve contextual information from earlier reasoning steps via attention mechanisms. We hypothesize that specific attention heads play a critical role: while most propagate alignment-consistent features supporting refusal, certain heads introduce competing signals that attenuate refusal-related representations, driving the observed score drop. Our detailed ablation experiments confirm this hypothesis, revealing a small set of **Refusal Suppression Heads**, sparsely distributed across deeper layers, that systematically reduce refusal scores. Removing these heads increases refusal scores at the thinking-end template and, in poorly aligned models, reduces attack success rates to below 10%.

To mitigate the Refusal Cliff, we propose a data filtering strategy that leverages internal representation signals to prioritize high-impact training samples. The key assumption is that effective safety fine-tuning should recover the model's early-stage refusal plateau – the stable region of refusal scores prior to suppression. We quantify misalignment between this plateau and the cliff position (where scores drop sharply) using a misalignment score, defined as the absolute difference between the plateau mean and the final-step score. We then fine-tune only on the most misaligned examples, targeting cases where refusal degradation is most severe. Using just the top 0.3% of samples, we reduce attack success rates on harmful-query benchmarks to below 5% while significantly lowering

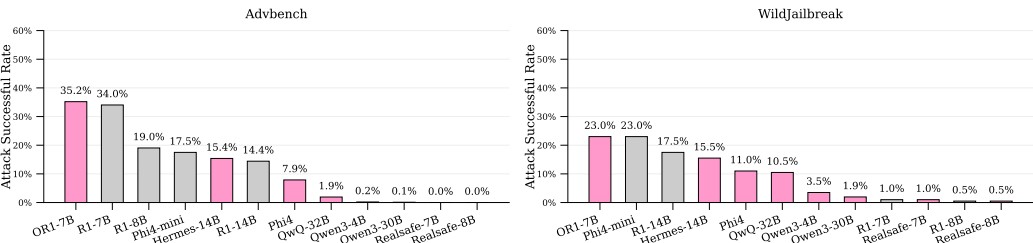

Figure 2: While some reasoning models achieve reasonable safety performance, a significant portion exhibit alarming vulnerabilities to adversarial attacks. We benchmark reasoning models (RLVR-based and Distillation-based) on AdvBench (Chao et al., 2024) and WildJailbreak (Jiang et al., 2024) with Attack Success Rate (ASR, the *lower* the *better*) as evaluation metric.

wall-clock training time relative to full-dataset fine-tuning. Compared to filtering methods such as LLM-as-a-judge (Gu et al., 2024), **Cliff-as-a-judge** achieves comparable safety gains with more flexible, metric-driven selection, demonstrating a clear *less-is-more* effect in alignment.

As summarized in Figure 1, our contributions are threefold:

- We identify and characterize the **Refusal Cliff**, a failure mode in which refusal intentions abruptly vanish at the reasoning output stage.
- We causally link this phenomenon to a small set of **Refusal Suppression Heads**, which undermine refusal behavior by suppressing alignment features.
- We introduce **Cliff-as-a-judge**, a probing-driven data selection method that mitigates safety vulnerabilities and achieves a *"less is more"* effect in safety alignment.

## 2 PRELIMINARIES

**Transformer.** We study reasoning models with Transformers (Vaswani et al., 2017) as a backbone. One Transformer model usually consists multiple of layers and an embedding layer. For an input $X_i \in \mathbb{R}^{n \times 1}$ with length $n$, it first passes through an embedding layer with hidden state size $d$, then passes all the Transformer layers:

$$H_i^{\text{att}} = H_i + \text{Attn}(\text{Norm}(X_i)), \ H_i = H_i^{\text{att}} + \text{MLP}(\text{Norm}(H_i^{\text{att}})). \tag{1}$$

Here, $H_i^{\text{att}}$ is the output hidden states of the attention block, and $H_i^{\text{mlp}}$ is the output of the MLP block for layer $i$.

**Models.** We evaluate two categories of reasoning models: *(i) RLVR-based models*, trained with *Reinforcement Learning with Verifiable Rewards (RLVR)* to enhance reasoning ability. We include QwQ (Team, 2025; Yang et al., 2024), Qwen3-Thinking (Yang et al., 2025), Skywork-OR1 (He et al., 2025), Phi-4-Reasoning (Abdin et al., 2025), and Hermes4 (Allan, 2018). *(ii) Distillation-based models*, trained by distilling reasoning traces from strong teacher models. We include DeepSeek-R1-Distill-Qwen-7B, DeepSeek-R1-Distill-LLaMA-8B, RealSafe-R1-7B, RealSafe-R1-8B (Zhang et al., 2025), and DeepSeek-R1-Distill-Qwen-14B (Guo et al., 2025). These selections cover diverse architectures, scales, and training paradigms. We assess safety using LlamaGuard-4 (Grattafiori et al., 2024), reporting *Attack Success Rate (ASR)*, defined as the fraction of harmful generations. As shown in Figure 2, safety alignment varies substantially across models: while some demonstrate robust alignment, others remain highly vulnerable.

**Datasets.** We evaluate safety using datasets that span both *vanilla attacks* – direct harmful queries – and *adversarial attacks* – crafted queries with deception and manipulation to bypass safeguards. For vanilla attacks, we use JailbreakBench (Chao et al., 2024), AdvBench (Zou et al., 2023b), and the vanilla subset of WildJailbreak (Jiang et al., 2024). For adversarial attacks, we use the adversarial subset of WildJailbreak.

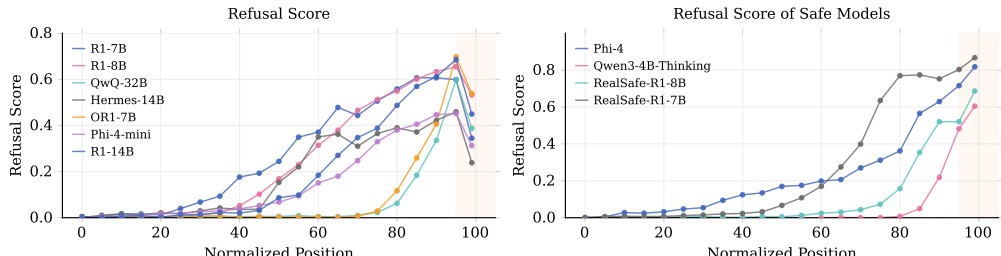

Figure 4: *Left:* Reasoning model with refusal cliff. We highlight the cliff position with orange background. *Right:* Well-aligned reasoning models experience no refusal cliff.

**Refusal Prober.** When an LLM encounters a harmful prompt, it will provide a refusal response to avoid giving users harmful information related to the question. Therefore, refusal examples *e.g., Sorry, I cannot...*, is a direct indicator for measuring the safety [1]. This also holds true for reasoning models. Recent work has shown that refusal behavior is often controlled by a single *refusal direction* within its activation space (Arditi et al., 2024). This direction is a vector that, when added to a hidden state, maximally increases the probability of generating a refusal. Due to this linear property, we can effectively identify this direction using a simple linear classifier *i.e.,* a refusal prober (Xu et al., 2024). The refusal prober is a logistic regression model that takes a hidden state vector $\boldsymbol{h}_j \in \boldsymbol{H}$ at token position $j$ as input and outputs the probability of refusal. The probability is calculated as:

$$P(\text{refusal}|\boldsymbol{h}_j) = \sigma(\boldsymbol{W}^T \boldsymbol{h}_j + b) \tag{2}$$

The prober is trained on a dataset with $N$ examples $\mathcal{D} = \{(\boldsymbol{h}_j^k, c^k)\}_{k=1}^N$ and the label $c$ is defined as:

$$c := \begin{cases} 1 & \text{for a refusal response (e.g., } Sorry, I cannot...), \\ 0 & \text{for a normal response (e.g., } The answer is...), \end{cases} \tag{3}$$

where $\boldsymbol{W} \in \mathbb{R}^{d \times 1}$ is the weight vector, $b$ is the bias, and $\sigma$ is the sigmoid function. We define the output probability as the **refusal score** of reasoning model at position $j$.

## 3 REFUSAL CLIFF IN REASONING MODELS

**Preparations.** We first train a refusal prober following the design in Equation 2. We trained the prober using the hidden states $\boldsymbol{h}$ extracted from the *final token position* in the last layer of each sequence in our dataset $\mathcal{D}$. For refusal response, we collect examples from Advbench (Zou et al., 2023b), and non-refusal response are collected from Ultrachatsft (Ding et al., 2023). The prober was trained for 5 epochs with 256 examples and achieved an average validation accuracy of over 95%. Loss curve and accuracy are shown in Figure 3. Considering the validation set is sampled from Advbench, as same as the examples source, we also test the Out of Distribution (OOD) accuracy of the prober on JailbreakBench (Chao et al., 2024). This high accuracy con-

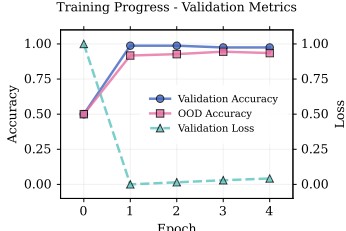

Figure 3: The loss, validation accuracy and OOD validation accuracy of the refusal prober.

firms that refusal behavior can be reliably predicted from a linear analysis of the model's internal states. Further details on hyperparameters and experimental settings are available in Appendix A.

**Refusal Cliff in Reasoning Models.** We probe the hidden states of reasoning models using the trained refusal prober to estimate the *refusal score* (defined at Eq. 2) at each token position. Probing is conducted from the first token of the prompt until the end of the model's reasoning process. Since the reasoning length varies across questions, we normalize all scores to a 0–100 scale, where 0 corresponds to the beginning and 100 corresponds to the final token position. By analyzing the

---

[1]Basically, the refusal response rate is given by $(1 - \text{ASR})$ for harmful prompts *i.e.,* cases where the model either refuses or responds harmfully. Although cases such as fake refusals or ambiguous answers exist, we do not analyze these kinds of complex behavior. We believe that a good model should either provide a clear refusal or a helpful answer.

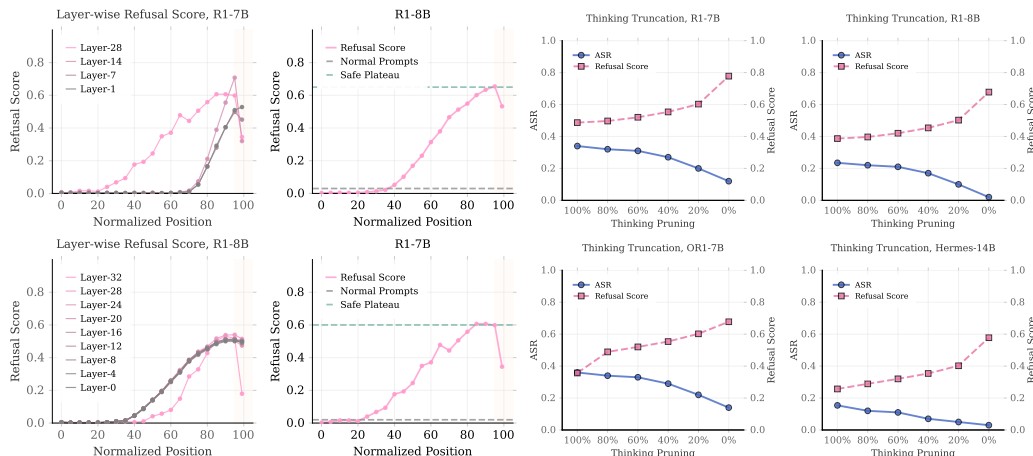

Figure 5: *The first column on the left:* Layer-wise refusal score of R1-Distill-Qwen-7B and R1-Distill-LLaMA-8B from shallow layers to deeper layers . *The second column on the left:* Comparison of refusal score in normal prompts and plateau values. Gray line is the average refusal score in normal prompts and Green line is the plateau of well-aligned family models. *The third and fourth column on the left:* Relation between thinking length and misalignment. We gradually clip thinking and force the model to directly answer.

refusal score of reasoning models, we can take a close look at their inner intention of tackling harmful requests. Results are illustrated in Figure 4 where the *left* are poorly aligned reasoning models and *right* are models that perform relatively well on safety benchmarks. Interestingly, for reasoning models that perform poorly on safety-related benchmarks, we observe a phenomenon we refer to as **Refusal Cliff**. As illustrated in Figure 4, the refusal score exhibits a gradual upward trend followed by a plateau phase. Critically, there is an abrupt decline in refusal scores at the terminal token positions, indicating that the model's internal intention transitions from rejecting the harmful request to complying with it.

**Properties.** To analyze the properties of the refusal cliff, we further conduct several experiments as shown in Figure 5. The refusal cliff exhibits four key properties and are summarized as below:

• The cliff is highly localized to the final few tokens of the reasoning process (as shown in the gray location in Figure 4), immediately preceding the model's output *i.e.,* the template region. In contrast, safety-aligned models such as Phi (Abdin et al., 2024) and Qwen3-thinking (Yang et al., 2025) show little to no cliff at such positions; their refusal scores may even increase as they conclude their reasoning.

• As shown in *the first column on the left*, Figure 5, The phenomenon is amplified in deeper layers, where the magnitude of the cliff increases substantially. Within deeper layers, the subsequent degradation in refusal efficacy becomes markedly more severe.

• As shown in *the second column on the left*, Figure 5, the cliff is preceded by a plateau, indicating that the model *recognizes* the harmful nature of the prompt despite its *eventual non-compliance*. During this plateau, the model's refusal intention is comparable to that of well-aligned variants.

• The model's thinking is vital for the refusal cliff. As we clip the thinking and directly prefilling the thinking end token *i.e.,* the thinking clipping operation (Jiang et al., 2025), to stop the thinking of the model in *the third and fourth column on the left*, Figure 5, we observe a lower level of refusal cliff and an increase of refusal response rate at the output.

## 4 WHO IS THE DEVIL IN REFUSAL CLIFF? A MECHANISTIC EXPLANATION FROM ATTENTION HEADS

We probe the refusal intention in reasoning models, discover refusal cliff, and discuss its properties. Since we know the refusal cliff exists, understanding how it happens is of great benefit to the safety and future improvements of reasoning models. In this section, we try to find out why.

## 4.1 Attention Heads in Refusal Cliff

**Why Attention Heads?** Intuitively, analyzing the phenomenon at the granularity of attention heads is natural: from a mechanistic interpretability perspective, attention heads are the main carriers of information routing in Transformer architectures, and different heads often specialize in diverse functions (Yin & Steinhardt, 2025; Olsson et al., 2022; Wu et al., 2024). It is also proven that attention heads play a key role in safety (Zhou et al., 2025b). In our case, the final tokens before the output closure template tokens *e.g.,* \n</think>\n\n, are strongly stereotyped between generations. However, for both the same template, refusal cliff happens in harmful examples but not benign ones. Therefore, a sudden disruption of this pattern during a refusal cliff suggests that certain heads have attended to specific prior content that triggers a mode change in the model.

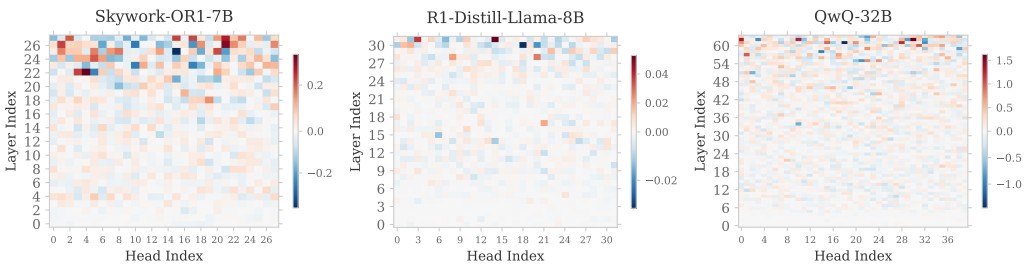

Figure 6: We trace the contributions of attention heads with probing. Red means the head contribute *positively* to final refusal and Blue indicates that this head contributes *negatively*.

**Tracing Attention Heads with Probing.** To accurately assess the causal impact of each attention head on refusal behavior, we employ a direct probing method to trace each head's contribution. Our approach is to individually evaluate the influence of each head's output at $t_{\text{cliff}}$, where the refusal cliff occurs. Specifically, for an attention head $h$ in any layer $i$, we first isolate its output vector $\boldsymbol{o}_{i,h,t_{\text{cliff}}}$. Following the standard Transformer architecture, this vector is projected into the residual stream via the attention block's output weight matrix, $\boldsymbol{W}_{O,i}$. To analyze the contribution of head $h$ alone, we construct a hypothetical residual update vector, $\Delta\boldsymbol{h}_{i,h,t_{\text{cliff}}}$, where only the output of head $h$ is active, while the outputs of all other heads in the same layer are zeroed out. Subsequently, we feed this vector containing the contribution of only a single head, $\Delta\boldsymbol{h}_{i,h,t_{\text{cliff}}}$, as input to our pretrained refusal prober to evaluate the head's independent refusal score. Its contribution score, $s_{i,h}$, is calculated as follows:

$$s_{i,h} = \boldsymbol{W}^T \Delta\boldsymbol{h}_{i,h,t_{\text{cliff}}} + b \tag{4}$$

where $\boldsymbol{W}$ and $b$ are the parameters of the prober (Eq. 2). We remove the sigmoid function so that we can directly trace the contribution of each attention head via logits (Heimersheim & Nanda, 2024; Zhang & Nanda, 2023). This score, $s_{i,h}$, directly quantifies the strength with which a single attention head, acting in isolation, pushes the model towards refusal or compliance. A score close to 1 indicates that the head promotes refusal, whereas a score close to 0 implies that it suppresses refusal.

**Tracing Results.** We aggregate the changes in refusal score for each head and visualize the results in Figure 6. In the heatmap, ref indicates a positive contribution to refusal behavior (*i.e.,* the head writes into the residual stream in a way that increases the refusal score for harmful prompts), while blue denotes a negative contribution (*i.e.,* the head decreases the refusal score, making refusals less likely). Notably, the contribution pattern is highly sparse: a small fraction of heads exhibit a strong negative correlation with refusal behavior, which we term the **Refusal Suppression Heads** [2].

## 4.2 Refusal Suppression Head Ablation

**Ablation Methodology.** We perform head ablation to *(i) cross-validate* the importance of the heads identified through tracing in the previous subsection and *(ii)* explore as a potential solution to tackle the unsatisfying safety alignment in reasoning models. Following previous work (Liu et al.,

---

[2]This definition is intended as a soft formulation, and in a later section we introduce a small threshold to facilitate the ablation analysis.

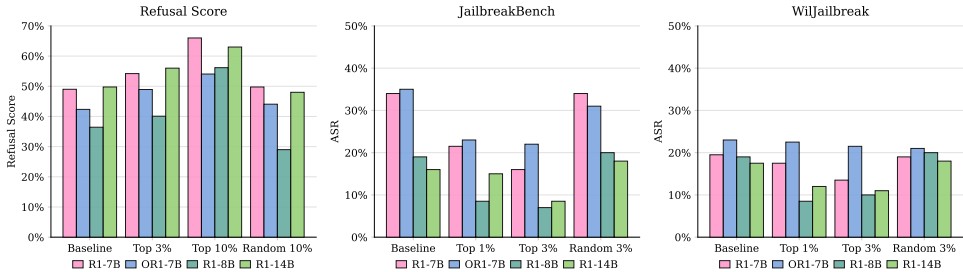

Figure 8: Results of ablating Refusal Suppression Heads. *Left:* The refusal score of prober output *i.e., the higher the better. Center:* The Attack Successful Rate (ASR) of JailbreakBench *i.e., the lower the better. Right:* The ASR of WildJailbreak.

2025b), we ablate attention heads one by one and evaluate the resulting changes in both the refusal score and the overall safety performance. We employ a scaling-down ablation, in which we introduce a scaling factor $\gamma$ to the output of the selected attention head to get the output $O$:

$$O = (\frac{QK^\top}{\sqrt{d}} \odot M) \cdot \gamma \cdot V, \text{ where } Q, K, V, O \in \mathbb{R}^{l \times d}, \ M \in \mathbb{R}^{l \times l}. \quad (5)$$

Here, $Q, K, V$ denote the query, key, and value matrices for this attention head, and $M$ is the causal mask used in decoder-only Transformers. When $\gamma = 0$, the output of that head is completely ablated, while $\gamma > 1$ amplifies the behavior of the original model. We also perform a renormalization method to keep the output norm stable and prevent generation collapse, following Zhang et al. (2024).

**Experiments.** We evaluated our method on JailbreakBench (vanilla attack) and WildJailbreak (adversarial attack) . We test the model performance with ablation on two level: *(i) Representation-level:* The refusal score of the prober after the ablation at the last token position on JailbreakBench. *(ii) Output-level:* The final Attack Successful Rate after the generation. We defined three thresholds, 1%, 3% and 10% [3], as criteria for identifying Refusal Suppression Heads, and set their contributions to zero using the scaling method described in Equation 5. Figure 6 presents the ablation results for these Refusal Suppression Heads. Our findings reveal that ablating as few as 10% of the identified attention heads can more than double the refusal score, while ablating only 3% of them is sufficient to reduce the probability of producing harmful outputs to below 10%.

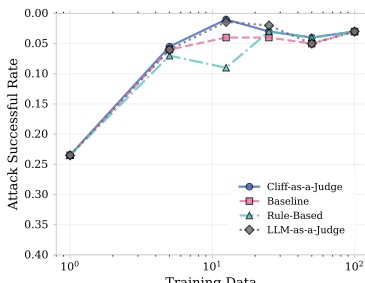

Figure 7: The pareto front beteen Examples and ASR.

**Limitations of Ablation.** We have proposed a seemingly practical solution for tackling the refusal cliff in reasoning models. However, we acknowledge that some readers may remain unconvinced by our conclusions—and rightly so. Intervention-based approaches are not perfect and suffer from several drawbacks: *(i)* The *superposition* of language model activations (Gao et al., 2024) *i.e.,* a single activation vector can be expressed as a linear combination of multiple sub-directions corresponding to different task domains, makes it difficult to intervene safely without compromising performance in other domains. *(ii)* Language models are capable of *self-repair* (Rushing & Nanda, 2024), which further limits the effectiveness of ablation alone in achieving optimal results. *(iii)* Intervening in a model's internal components requires redesigned infrastructure and cannot be readily applied to existing tools. We will present a more practical approach in the next section.

---

[3]We use 1% and 3% for generation (and 3% and 10% for refusal score) because ablating a large number of heads may lead to generation collapse.

Table 2: Benchmark results on safety-related tasks and reasoning-related tasks.

| Method | Examples | JailbreakBench | | WildJailbreak | | MMLU-Pro | | ARC-C | |
| | | *Metric: ASR($\downarrow$)* | | | | *Metric: Acc($\uparrow$)* | | | |
| | | R1-8B | R1-7B | R1-8B | R1-7B | R1-8B | R1-7B | R1-8B | R1-7B |
|---|---|---|---|---|---|---|---|---|---|
| Baseline | - | 32.0% | 31.2% | 19.0% | 38.0% | 42.7% | 45.4% | 40.7% | 41.8% |
| Full Training Dataset | 40k | 2.5% | 1.0% | 2.0% | 1.0% | 40.7% | 42.9% | 41.3% | 39.9% |
| Rule-Based | 21k$^{-46.1\%}$ | 1.0% | 2.5% | 5.2% | 2.4% | 40.8% | 43.4% | 40.9% | 40.8% |
| LLM-as-a-Judge | 5.6k$^{-86.0\%}$ | 4.0% | 1.5% | 6.5% | 1.8% | 40.8% | 43.7% | 40.5% | 40.4% |
| **Cliff-as-a-judge** | 700$^{-98.3\%}$ | 5.0% | 3.0% | 6.0% | 6.0% | 41.7% | 44.7% | 41.4% | 41.2% |

## 5  CLIFF-AS-A-JUDGE: EFFICIENT ALIGNMENT VIA DATA SELECTION

### 5.1  METHODOLOGY

**Motivations.**  From our previous experiments, it is evident that a misaligned reasoning model is not inherently incapable of safe behavior. On the contrary, such a model can often correctly identify the harmful nature of a prompt and internally reflect an intention to refuse during its reasoning process. Under this hypothesis, it follows that aligning an unsafe reasoning model may require only a small set of high-quality alignment examples, thereby achieving a *less-is-more* effect (Zhou et al., 2023).

**Cliff-as-a-judge.**  We propose a cliff-based data selection method. Formally, given a dataset $D$ and a budget $k$, data selection is to get an optimal subset $S \subset D$, $|S| = k$ to optimize its alignment performance. Specifically, suppose that, for a given sample, the

Table 1: Comparison between data selection methods.

| Method | Continuous | Judge Model | Performance |
|---|---|---|---|
| Baseline | N/A | None | Good |
| Rule-Based | False | None | Moderate |
| LLM-as-a-judge | False | LLM ($>$1B) | Good |
| **Cliff-as-a-judge** | True | Prober ($<$1M) | Good |

model's maximum refusal intention *i.e.,* the plateau, expressed within its internal thinking corresponds to a probed refusal score $I$, and its final generated refusal score is $I'$ after any cliff drop or suppression. We define the misalignment score $\text{MS} = I - I'$ as a measure of how much the refusal intention expressed in internal reasoning is suppressed in the final output. Intuitively, the most effective subset of alignment data consists of samples with the highest misalignment scores, where training on this data can most efficiently repair safety alignment. Therefore, the optimal selection via Cliff-as-a-judge is:

$$\theta^* = \arg\min_\theta \ \mathcal{L}_{\text{align}} \left( \arg\max_{S \subset D, |S|=k} \ \frac{1}{k} \sum_{x \in S} \text{MS}(x) \ ; \ \theta \right) \quad (6)$$

where $\mathcal{L}_{\text{align}}$ denotes an alignment-oriented objective (*e.g.,* Attack Successful Rate). We compare our method with other baselines in Table 1, where Cliff-as-a-judge provides a continuous metric, allows flexible selection of the number of examples, employs a lightweight judge model, and achieves strong performance.

### 5.2  EXPERIMENTS

**Baselines.**  We adopt the training set from WildJailbreak (Jiang et al., 2024) as our safety alignment corpus with 40k examples. This dataset contains both standard (vanilla) jailbreak attacks and more challenging adversarial jailbreak cases. For baseline data selection methods, we consider: *(i)* full-example training (*i.e.,* the unfiltered baseline), *(ii)* rule-based selection (Liu et al., 2025b; Lab et al., 2025), where unsafe cases are identified using keyword matching, *(iii)* LLM-as-a-judge (Gu et al., 2024; Lambert et al., 2024; Zhang et al., 2025), where using LlamaGuard (Grattafiori et al., 2024). We also select MMLU-Pro (Wang et al., 2024) and ARC-Challenge (Clark et al., 2018) to benchmark the reasoning ability after alignment.

**Experimental Results.**  We perform safety fine-tuning on our selected datasets. Table 2 demonstrates the effectiveness of our Cliff-as-a-judge data selection method across three safety benchmarks. While the baseline models exhibit concerning vulnerabilities with ASR of 19.0-38.0%,

training on the full dataset reduces ASR to 1.0-2.5%. Remarkably, our method achieves comparable safety performance using only 700 examples (98.3% data reduction). This substantially outperforms other filtering approaches: Rule-based selection requires 21,566 examples (-46.1%) and LLM-as-a-judge needs 5,616 examples (-86.0%) to achieve similar results. As shown in Figure 7's Pareto frontier analysis, our approach optimally balances data efficiency with safety performance, translating to reduction in training time while maintaining effective safety alignment across different model architectures. Also, our benchmarking on MMLU-Pro and ARC-C demonstrates that Cliff-as-a-judge is most effective in preserving the model's original reasoning capabilities, while requiring fewer yet higher-quality examples.

## 6 RELATED WORKS

**Safety of Large Reasoning Model.** The development of reasoning models extends safety beyond direct harmfulness classification to deliberate, step-by-step judgment (Wang et al., 2025) with robustness to jailbreak attempts (Zaremba et al., 2025; Kim et al., 2025). However, studies also show that this generalization is fragile and can be exploited (Kuo et al., 2025; Yan et al., 2025; Zheng et al., 2025; Jiang et al., 2025). In response, recent work proposes frameworks—both by evaluating and mitigating risks within reasoning traces themselves (Li et al., 2025b; Zheng et al., 2025) and by improving safer outputs (Zhu et al., 2025; Jiang et al., 2025). From a different angle, we investigate the mechanistic roots of LRMs' safety vulnerabilities and offer insights for future solutions.

**Mechanistic Interpretability for LLM Safety.** Mechanistic Interpretability (MI) seeks to reverse-engineer specific model behaviors and functions so their internal mechanisms become human-understandable. Research in this area spans multiple granularities: individual neurons (Gurnee et al., 2023; Stolfo et al., 2024), representations (Marks & Tegmark, 2024; Gurnee & Tegmark, 2024), and larger functional units like MLP (Geva et al., 2021; 2022) and attention heads (McDougall et al., 2023; Gould et al., 2024). Building on these foundations, MI has been increasingly applied to LLM safety (Bereska & Gavves, 2024). One thread focuses on representation-level analyses of safety behavior and on techniques for editing safety-related representations (Leong et al., 2023; Zou et al., 2023a; Arditi et al., 2024; Cao et al., 2025; Lee et al., 2025a; Li et al., 2025c; Shen et al., 2025; Xu et al., 2024; Lee et al., 2025b). Another examines components directly implicated in safety, including neurons (Zhao et al., 2025), attention heads (Zhu et al., 2024; Zhou et al., 2025b), and MLPs (Lee et al., 2024; Luo et al., 2024). Complementary work studies safety-relevant parameters themselves (Wei et al., 2024; Yi et al., 2025; Gu et al., 2025). A parallel line of progress decomposes representations into interpretable, sparse features, enabling automated explanations of safety mechanisms (Minder et al., 2025). These methods suggest promising avenues for achieving more robust safety alignment at the representation level (Liu et al., 2024; Zou et al., 2024; Rosati et al., 2024; Yin et al., 2025).

## 7 LIMITATIONS

While our study sheds light on the mechanistic roots and offers mitigation strategies, several limitations remain. First, our mechanistic analysis focuses primarily on attention heads, leaving other architectural components such as MLP blocks, positional encodings, and cross-layer interactions underexplored. Second, our data-recipe method depends on having access to the model's internal representations and refusal scores, which is feasible for open models but may be impractical for proprietary systems. Investigation of proxy metrics or black-box analogues remains future work.

## 8 CONCLUSIONS

In this work, we identified and mechanistically characterized a novel safety failure in large reasoning models – the **refusal cliff**. Through causal tracing, we discovered a small set of Refusal Suppression Heads whose negative contributions are responsible for this phenomenon. Targeted ablation of these heads significantly improves refusal rates, confirming their causal role. Building on these findings, we proposed a targeted safety fine-tuning data recipe that selects training examples most susceptible to the refusal cliff. Our experiments show that these methods can improve safety alignment with minimal performance trade-offs while reducing the training cost.

ETHICS STATEMENT

Our research aims to enhance the safety and reliability of Large Reasoning Models (LRMs) by identifying and mitigating a critical failure mode, the "Refusal Cliff." We believe this work contributes positively to the responsible development of AI. However, we acknowledge several ethical considerations inherent in this line of research. Our work involves the analysis of model vulnerabilities to harmful and malicious prompts, which carries a potential dual-use risk. To mitigate this, we have focused on revealing the underlying *mechanisms* of failure rather than developing novel, easily replicable jailbreak techniques. Our proposed solution, "Cliff-as-a-Judge," is a defensive data selection strategy designed to strengthen model safety. The datasets used, such as AdvBench and WildJailbreak, are established benchmarks and were used strictly for evaluating and improving model refusal capabilities without generating new harmful content. We believe our findings can help improve the alignment of models to reduce harmful or biased outputs and encourage the community to build upon our mechanistic insights to develop more robust and ethically aligned AI systems. All research was conducted in adherence to the ICLR Code of Ethics.

REPRODUCIBILITY STATEMENT

We are committed to ensuring the reproducibility of our research. All models used in our experiments (e.g., from the Qwen, DeepSeek, Skywork, and Phi families) and datasets (e.g., AdvBench, JailbreakBench, and WildJailbreak) are publicly available and detailed in Section 2. The implementation details for our core methodologies are provided in the appendix. Specifically, Appendix A contains the complete setup for training the refusal prober, including hyperparameters and data preprocessing. The procedures for attention head tracing (Section 4), head ablation (Section 4), and the fine-tuning process for our "Cliff-as-a-Judge" method (Section 5) are described with sufficient detail for replication. To further facilitate reproducibility, we will release our source code, which includes scripts for data processing, prober training, causal analysis, and model fine-tuning, upon publication of this paper.

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

## USE OF LLMS

In the preparation of this manuscript, we utilized LLMs as a writing assistant. The use of LLMs was confined to tasks such as improving grammar, refining phrasing for clarity, and polishing the overall language of the paper. All core scientific contributions, including the conceptualization of ideas, the design and execution of experiments, the analysis of results, and the conclusions, are entirely the work of the human authors. The authors bear full responsibility for the content and claims presented in this work.

## A    PROBER

In this section, we provide a detailed description of the architecture, data collection, and training procedure for the refusal prober used in our experiments. This prober is a linear classifier designed to predict whether a model will refuse a harmful request based on its internal hidden states.

**Prober Architecture.**    The prober is implemented as a simple linear classifier. Given a hidden state vector $h \in \mathbb{R}^d$ from the reasoning model, where $d$ is the hidden dimension size, the prober computes a single logit. This is followed by a sigmoid function to produce a refusal probability, as defined in Equation 2. The model is implemented in PyTorch using a single 'torch.nn.Linear' layer. We use the Binary Cross-Entropy with Logits loss function ('nn.BCEWithLogitsLoss') for training, which is numerically stable and suitable for binary classification tasks.

**Dataset Collection and Preprocessing.**    To train the prober, we constructed a balanced dataset of hidden states corresponding to both refusal and non-refusal responses.

- **Refusal Examples (Positive Class):** We collected examples where the model refused to comply with a harmful prompt. These were sourced from the AdvBench dataset (Zou et al., 2023b). An output was labeled as a refusal if it contained keywords like "I'm sorry," "I cannot," or similar phrases within the first 32 tokens of the response.
- **Non-Refusal Examples (Negative Class):** For the non-refusal class, we used harmless prompts and their corresponding compliant answers from the UltraChat-SFT dataset (Ding et al., 2023).

For each example in both classes, we fed the full input sequence (user prompt + model's chain of thought + thinking-end template) into the target reasoning model. We then extracted the hidden state vector from the \*\*final token position\*\* at the \*\*last transformer layer\*\*. These hidden state vectors form the training data for our prober.

**Training Details.** The prober was trained on the collected hidden states. Before training, we balanced the dataset by randomly downsampling the larger class to match the number of samples in the smaller class, ensuring an equal number of refusal and non-refusal examples. The full dataset was then split into training (80%) and validation (20%) sets.

The training hyperparameters are as follows:

- **Optimizer:** Adam
- **Learning Rate:** $1 \times 10^{-3}$
- **Batch Size:** 256
- **Epochs:** 5

We selected the model checkpoint that achieved the highest accuracy on the validation set. As reported in Section 3, the final prober achieved over 95% validation accuracy on in-distribution data and demonstrated strong generalization to an out-of-distribution (OOD) dataset, JailbreakBench. This high accuracy confirms that the prober reliably captures the model's refusal intention from its final hidden state.

## B  SUPERVISED FINE-TUNING DETAILS

We performed full-parameter supervised fine-tuning (SFT) to repair the safety alignment of the reasoning models using the data subsets selected by our Cliff-as-a-Judge method. The entire training process was conducted using the LLaMA-Factory library. The base model for the fine-tuning experiments reported in Section 5 was `deepseek-ai/DeepSeek-R1-Distill-Qwen-7B`. We utilized DeepSpeed ZeRO Stage 2 for efficient distributed training. The key hyperparameters and configuration settings are detailed below:

- **Finetuning Type:** Full-parameter SFT
- **Learning Rate:** $5 \times 10^{-6}$
- **LR Scheduler:** Linear
- **Epochs:** 1.0
- **Batch Size:** 1 per device with 4 gradient accumulation steps, resulting in an effective batch size of 4.
- **Optimizer:** AdamW (`adamw_torch`)
- **Precision:** BF16
- **Max Sequence Length:** 16,384
- **Attention Implementation:** Flash Attention
- **Prompt Template:** `deepseekr1`
- **Distributed Training:** DeepSpeed ZeRO Stage 2

