# OpenReview forum: "Refusal Falls off a Cliff: How Safety Alignment Fails in Reasoning?"
_ICLR.cc/2026/Conference — ICLR 2026 Conference Withdrawn Submission_

### Official Review · Reviewer_tcSd · 2025-10-20

**Soundness:** 1
**Presentation:** 2
**Contribution:** 1
**Rating:** 2
**Confidence:** 4

**Summary:**

Reasoning models seem to exhibit more safety vulnerabilities than non-reasoning models; that is, they tend to be less likely to refuse harmful requests. This paper investigates *why* reasoning models are less robust. The authors train a probe on model activations to predict refusal, and find that the probes scores are fairly strong throughout the reasoning trace, but abruptly drop ("fall off a cliff") at the end. The authors find that there is a small set of heads that suppress refusal. The authors additionally use the probe score as a way of efficiently selecting good datapoints to improve robustness, by selecting datapoints where the "refusal cliff" is most prominent.

**Strengths:**

- Cheap probe-based data selection
	- The final section suggests an idea to cheaply identify training data that would most effectively safety tune a model. It works by identifying samples with the largest "refusal cliff", as these are intuitively the most "problematic" samples, that the model could learn most from. Going from the "refusal cliff" observation to this data selection procedure is a nice, practical idea.

**Weaknesses:**

- Insufficient technical detail
	- I feel that there is insufficient technical detail for me to confidently believe in the results.
	- See the questions section for a list of my questions regarding technical details, some of which are critical to interpreting the main claims of the paper (e.g., the main claim of a "refusal cliff" is predicated on the refusal probe results, which are in turn predicated on the technical details of probe training and dataset construction).
- Probe design concerns
	- The probe is trained only on final token positions at the final layer, but then applied across all positions and potentially all layers. This distribution shift is not addressed, raising questions about what the probe actually measures at intermediate positions.
	- Training on the final layer at the final position likely captures something close to the output logit direction (e.g., "Sorry" vs "Sure" unembedding), which may not reflect genuine "refusal intention" at earlier reasoning steps.
- Shallow mechanistic analysis
	- Any direction that shows up in the residual stream must be written to by some model component (an attention head or MLP). It is thus not surprising to find a set of model components writing to the probe direction, or that ablating these model components weakens the probe direction. It is therefore unclear to me why the section on "refusal suppression heads" is included in the manuscript; I don't think it adds to any understanding of what's going on.
- Lacks statistical rigor
	- There are no error bars or confidence intervals in any of the results.
- Clarity and presentation issues
	- Figure 5's "thinking pruning" terminology is confusing (does 100% mean fully pruned or fully retained?).
	- Figure 8's grouping makes within-model comparisons difficult.
	- Unclear how to reconcile Figure 7 with Table 2.

**Questions:**

- What are the details of assessing the safety of generations?
	- The paper explains the LlamaGuard-4 is used. What text is given to LlamaGuard-4 for judgement? Is it just the final output (excluding reasoning)? Or is it the assistant's whole response (including reasoning)?
	- Also, in the paragraph "Refusal Prober", it suggests that you assess safety by searching for individual substrings like "Sorry, I cannot" (although details are not provided). I think having one standardized way of categorizing "refusal" vs "non-refusal" would make more sense.
- Refusal prober details
	- Why is the probe trained on the final token position?
		- The main experiment runs the probe over all token positions. Isn't this potentially very out of distribution for the probe? If the intention is to run the probe over all token positions, why not train it on activations from all token positions?
	- Why is the probe trained on the final layer?
		- The final layer likely contains most strongly information about the unembeddings. Since you're probing at the final token position, this likely corresponds to something like the "Sorry" unembedding direction minus the "Sure" unembedding direction (not sure about the exact tokens, since this will depend on what tokens the model prefers to start its refusal vs non-refusal responses with, but hopefully you get the point).
		- Why is this a reasonable thing to do?
- Dataset construction
	- What are the details of dataset construction?
		- For example, for each model that you train a probe for, do you generate completions from that model on your dataset of prompts, and then use those rollouts to train the probes for that model? Or do you do rollouts from a single model, and use that single model to train probes across all models? What are the details of generation? Given a reasoning trace, does the model consistently refuse or not refuse? Or is it probabilisitic?
		- Do you include any harmful prompts that are not refused?
	- For the JailbreakBench OOD evaluation, is this dataset balanced? I'd think that JailbreakBench only contains harmful prompts that should be refused. Is there also a harmless OOD dataset that you use to balance the total OOD set?
- Figure clarifications
	- In Figure 5, what does "thinking pruning" mean? Does 100% thinking pruning mean that 100% the reasoning has been pruned, and so there is no reasoning left? Or does it mean that 100% of the reasoning remains (e.g., nothing has been pruned)?
- Attention head interventions
	- In Section 4.2, at which token position(s) is the intervention performed?
		- Consider changing the grouping of bars in Figure 8 so that each group is the same model, and the colors differentiate the intervention. We ideally want to compare the interventions within each model.
	- In equation 4, why include the bias term?
		- I believe the bias should not be included, as you are interested in whether heads contribute positively or negatively; adding a bias can change this. E.g., if the bias is positive, then a negative head contribution might show up as positive.
		- It's also unclear how this score is "A score close to 1 indicates that the head promotes refusal, whereas a score close to 0 implies that it suppresses refusal." There's no reason for the result of equation 4 to be within 0-1. In fact, in Figure 6, they are clearly not.
- Cliff-as-a-judge experiments
	- How did you choose the number of examples for each method in Table 2?
	- How do you explain the discrepancy between Figure 7 and the claim that "our approach optimally balances data efficiency with safety performance"?
		- Figure 7 seems to show that LLM-as-a-judge is as good (or even better) than cliff-as-a-judge.

---

### Official Review · Reviewer_1dk4 · 2025-10-29

**Soundness:** 3
**Presentation:** 3
**Contribution:** 2
**Rating:** 4
**Confidence:** 4

**Summary:**

This paper investigates an interesting and important phenomenon that reasoning models' refusal intention abruptly falls the cliff before generating the final output. This case is related to previous work like llm faithfulness[1][2] and model internal representation[3][4]. Through causal intervention, the authors figure out a set of “Refusal Suppression Heads” responsible for the cliff. Building on this insight, they propose Cliff-as-a-Judge, a data selection method that prioritizes training examples with the largest refusal cliffs, achieving strong safety improvements with only 1.7% of standard safety data.

**Strengths:**

1. The investigated question is truly important and realistic, which makes the motivation strong and meaningful.

2. The previous rethinking part is meticulous and convincing with detailed ablation studies.

3. The data selection method to tackle this problem is impressive, with fewer data to achieve a better defense performance.

4. The authors include different types and sizes reasoning models in the rethinking part, which convinces the experimental results of this paper.

5. The presentation is good and easy to follow.

**Weaknesses:**

1. This paper doesn't investigate the over-refusal problem after data selection. It's unclear if the low ASR comes from over-refusal to input questions.

2. Human evaluation is lacked. The experiment ASR is measured by LLaMA-Guard-4, however, the human evaluation is lacked. It's very important to include human evaluation to reduce the False Positive Rate and False Negative Rate. This makes the ASR reported in the article less convincing.

3. I really like the probing part (Lines 194-206) that the author separates the problem from an in distribution(trained on vanilla) and out of distribution(test on adversarial jailbreak) aspect. However, the main method of this paper: CLIFF-AS-A-JUDGE utilizes both "vanilla attacks and more challenging adversarial jailbreak cases" as the training dataset (Lines 421-423), which mixes the in distribution and out of distribution setting. Therefore, the testing results in Table 2 seems to become an in distribution test. Out of distribution test is truly important in safety alignment because attacks are often unseen and unknown[1][2][3]. Therefore I strongly suggest the authors to include
at least some unseen attacks like GCG, Simple Adaptive Attacks, Code Attacks to demonstrate the generalization ability of their method, or separate clearly the in distribution and out of distribution settings in training and testing time.

4. The Cliff-as-a-Judge method requires access to internal hidden states, so it's a white box setting. However, this paper mainly include black-box attacks. It remains unclear if it's also effective on white box attacks.

5. It's OK to test with MMLU-Pro and ARC-C to show the reasoning ability, but the general ability like if-eval, Alpaca-eval, the hard math ability like MATH is also important for a reasoning model.

6. The investigated problem is similar to the unfaithfulness[4] and hallucination[5] problem of Large Reasoning Models. So please include some discussion on the similarity and difference between them.

[1] Improving Alignment and Robustness with Circuit Breakers

[2] Refuse Whenever You Feel Unsafe: Improving Safety in LLMs via Decoupled Refusal Training

[3] Safety Reasoning with Guidelines

[4] Reasoning Models Don't Always Say What They Think

[5] Are Reasoning Models More Prone to Hallucination?

**Questions:**

1. How do you see the "REFUSAL FALLS OFF A CLIFF" phenomenon's connection to the unfaithfulness and hallucination of LRMs?

---

### Official Review · Reviewer_K119 · 2025-10-31

**Soundness:** 2
**Presentation:** 3
**Contribution:** 2
**Rating:** 4
**Confidence:** 4

**Summary:**

This paper investigates why safety alignment often fails in Large Reasoning Models (LRMs) by introducing a novel phenomenon called the Refusal Cliff—where models maintain strong refusal intentions throughout the reasoning process but suddenly lose them at the final output stage.
They propose Cliff-as-a-Judge, a data selection method that leverages internal representation signals to select high-impact alignment samples, achieving comparable safety improvement with only 1.7% of standard training data. The work aims to offer both mechanistic insight and practical efficiency in safety alignment for reasoning models.

**Strengths:**

1. The paper provides a clear, mechanistic explanation for why LRMs sometimes fail to refuse harmful prompts despite seemingly correct internal reasoning. The identification of Refusal Suppression Heads offers a concrete interpretability advance.

2. The proposed Cliff-as-a-Judge achieves significant safety improvement with minimal fine-tuning data, showing cost-effectiveness and potential for scalable safety alignment.

**Weaknesses:**

1. The paper does not clearly justify why the proposed approach is necessary, given that many recent reasoning-aligned models (e.g.,  Qwen3-30B, RealSafe) already achieve near-perfect refusal rates. The incremental need over existing safety-aligned models is not well articulated.

2. Only two benchmarks (JailbreakBench and WildJailbreak) are used, which are relatively simple. Missing evaluations on over-refusal (false positive safety behavior) and generalization benchmarks such as WildChat or OR-Bench.

3. The paper should include a direct comparison to STAR-1: Safer Alignment of Reasoning LLMs with 1K Data, which also claims efficient safety alignment with small datasets. Without this, the “less-is-more” claim remains weakly supported.

4. The paper’s flow is somewhat confusing, the method section appears too late, and the transitions between probing analysis and data-selection pipeline could be reorganized for clarity.

**Questions:**

1. How would the proposed approach work for closed-source reasoning models (e.g., GPT, Gemini), where internal representations cannot be accessed?

2. Since the probe can accurately estimate refusal behavior, could it be integrated into reinforcement learning frameworks to guide alignment dynamically rather than only during SFT?

3. Why was STAR-1 excluded as a baseline, given that it also focuses on aligning reasoning LLMs with minimal data (1K samples)?

4. Has the method been tested on harmless prompts to ensure it does not increase over-refusal or degrade helpfulness?

5. Considering some aligned models already achieve almost perfect safety, in what scenarios would Cliff-as-a-Judge still be necessary or beneficial?

---

### Official Review · Reviewer_jYPS · 2025-11-02

**Soundness:** 2
**Presentation:** 2
**Contribution:** 2
**Rating:** 4
**Confidence:** 4

**Summary:**

This work introduces a safety-specific vulnerability in reasoning models — the **refusal cliff**. The phenomenon describes how, during generation, a model may reverse its refusal behavior just before the final few tokens, resulting in failure to reject risky prompts that should have been refused. Based on this observation, the authors further propose a mechanism-driven, efficient refusal data selection approach: by incorporating samples with significant refusal cliff characteristics into training, they achieve better refusal effectiveness, while maintaining stable performance on MMLU and ARC.

**Strengths:**

1. Defines an intriguing phenomenon — **refusal cliff** — and proposes a novel explanatory mechanism for the stability (or instability) of model outputs.
2. Uses causal intervention to identify a sparsely distributed set of attention heads in the Transformer (only about 3% of heads) that negatively regulate refusal behavior in deep layers.
3. Proposes an efficient method for generating safety-related training data: leverages the refusal cliff to selectively choose data, achieving near-baseline safety performance using only 10% of the data — without performance degradation on MMLU and ARC.

**Weaknesses:**

1. Although the authors initially pose the research question *“What mechanism makes the safety alignment vulnerable in reasoning models?”*, the premise of this question is debatable. Many prior studies have found that safety alignment in reasoning models is actually *more* robust than in non-reasoning models. The claimed vulnerability seems more related to training data issues than an inherent mechanism. To argue for a mechanism-level cause, more experiments are needed to exclude data-related effects.
2. The assumption that *refusal ≈ safety* is insufficient; more clarification or a scope limitation is needed. Similarly, using a judger that detects refusal only by matching keywords like “I can’t” risks substantial false positives.
3. Using linear probes to predict refusal behavior may overlook complex non-linear interactions. Although linear models have strong interpretability, the actual refusal mechanism may involve high-dimensional features; thus, a linear probe can only reflect correlation, not conclusively prove the existence of a “refusal intent.” High probe scores may merely indicate harmfulness detection rather than the actual refusal decision.

-  **Typos**
1.  Line 41 zhang2025realsafe
2. Line 43， invaluable insights？

**Questions:**

1. Current experiments are insufficient. It remains unverified whether the refusal cliff inevitably occurs under different training methods, reasoning chain lengths, and safety paradigms. The set of training methods, baseline models, and benchmarks is small. The authors should also evaluate on over-refusal benchmarks (e.g., XSTest) to check if their method introduces excessive refusals.
2. The authors mention the “superposition” phenomenon (a single activation vector may contain multiple sub-directions). Is there a more theoretical way to explain why the experimental results are not affected by this?

---

### Note · Authors · 2025-11-21

I have read and agree with the venue's withdrawal policy on behalf of myself and my co-authors.